# Chemical Supercritical Fluid Infiltration of Pyrocarbon with Thermal Gradients: Deposition Kinetics and Multiphysics Modeling

**Gerard L. Vignoles ***[ID]**, Gaëtan Talué, Quentin Badey, Alain Guette, René Pailler, Yann Le Petitcorps and Laurence Maillé**

Laboratory for ThermoStructural Composites (LCTS), University Bordeaux, CNRS, Safran, CEA, F33600 Pessac, France; gaetan.talue@lcts.u-bordeaux.fr (G.T.); badey@lcts.u-bordeaux.fr (Q.B.); alguette@free.fr (A.G.); famille.pailler@wanadoo.fr (R.P.); yann.le-petitcorps@u-bordeaux.fr (Y.L.P.); maille@lcts.u-bordeaux.fr (L.M.)
\* Correspondence: vinhola@lcts.u-bordeaux.fr

**Abstract:** The chemical supercritical fluid infiltration process is a recent variation of the chemical vapor infiltration (CVI) process that allows rapid and efficient manufacturing of ceramic-matrix composites (CMCs), albeit still needing optimization. This article proposes a quantitative assessment of the process dynamics through experiments and modeling. The kinetics of carbon deposition were determined through two sets of experiments: CVD on a single filament at pressures between 10 and 50 bar and infiltration at pressures ranging between 50 and 120 bar. The CVI experiments were conducted under important thermal gradients and were interpreted using a model-based reconstitution of these gradients. We found that (i) the kinetic law has to incorporate the potential effect of the reverse reaction (i.e., etching of C by $H_2$); (ii) the activation energy and pre-exponential factor both decrease with pressure up to 50 bar, then remain roughly constant, and (iii) although the apparent activation energy is modest, a favorable situation occurs in which an infiltration front builds up and travels from the hottest to the coldest part of the preform due to the presence of sufficient heat flux. A numerical simulation of the process, based on the solution of momentum, heat, and mass balance equations, fed with appropriate laws for the effective transfer properties of the porous medium and their evolution with infiltration progress, was performed and validated by comparing the simulated and actual infiltration profiles.

**Keywords:** process modeling; heat and mass transfer; supercritical fluid; infiltration front; chemical vapor infiltration (CVI); ceramic-matrix composites (CMCs)

## 1. Introduction

The chemical vapor infiltration (CVI) process is one of the routes used for the manufacturing of ceramic-matrix composites [1,2], among which SiC-matrix composites [3,4], ultra-high temperature ceramics (UHTCs) [5], and especially carbon-fiber-reinforced carbon ($C_f$/C) composites [6,7]. This process, a variation of chemical vapor deposition (CVD) [8], has the advantage of producing high-quality matrices without damaging to the fibers, thus allowing a nearly optimal performance of the constituents in the composite [9,10]. However, the baseline CVI process, isothermal and isobaric CVI (I-CVI), though economically efficient for $C_f$/C brake production, is lengthy and expensive to implement. Many variations of CVI were sought in order to speed up the infiltration rate [11]. The most explored methods feature the incorporation of a thermal gradient, obtained by various methods that can be collectively designated as TG-CVI [12]. Another method is to accelerate chemical deposition kinetics through the increase in the gaseous reactant partial pressure. This is made possible as soon as a sufficient thermal gradient is present, for example in the film-boiling process [13–16] in which the exterior part of the fibrous preform is in contact with the

boiling precursor, thereby maintaining a strong thermal gradient inside it. In this TG-CVI variation, the external pressure is ambient instead of the reduced pressure necessary for the correct infiltration in I-CVI conditions. Advancing the idea is to the use of very high pressures, resulting in the development of chemical super-critical fluid infiltration (CSCFI or SCFIn) [17]. Rapid infiltrations of pyrolytic carbon from methane [18] and of SiC from polycarbosilane [19] have been reported.

In order to optimize and upscale this process, modeling is an essential tool. Many studies have been conducted to achieve these goals [20]. For instance, models for the forced-flow CVI with temperature gradients [21], the CVI process with radio-frequency heating (RF-CVI) [22] and the 'film-boiling TG-CVI process [23] have been developed and validated experimentally. Depending on the precise setup and conditions, it has been shown that an infiltration front can appear and travel through the porous preform, achieving a correct inside-out infiltration [24,25]. This front requires a minimal thermal gradient to exist [26], a criterion that is not always obtained in practice [20].

The chemical system we considered here has also attracted attention due to its potential efficiency in achieving the clean conversion of methane, a greenhouse gas, into hydrogen, with the solid carbon being considered here only as a by-product [27]. For instance, a methane pyrolysis reactor was developed for $CO_2$ recycling in the International Space Station [28]. Increasing temperature and pressure were also shown to favor methane conversion in this situation.

The aim of the study presented here was to obtain necessary data and knowledge about the process to be able to scale it up. Two elements are of paramount importance to reach this goal: (i) obtaining sufficient information on the deposition/infiltration kinetics and (ii) setting up and validating a numerical model of the CSCFI process. This was achieved with the resolution by the finite elements (FEs) of heat, momentum, and mass balances; validation was conducted with experimental data, providing some insight into the process. In particular, it was necessary to verify that the temperature gradient was steep enough to favor the existence of an inside-out infiltration regime and to obtain details on other non-measured quantities during the process.

The paper is organized as follows: First, the process and the model setup are briefly described, then, experimental deposition and infiltration results are presented and discussed. The next section is devoted to the interpretation of the infiltration experiments through numerical simulation. The study first involved a reconstitution of the temperature and pressure conditions present in the reactor; second, analytical tools were used for estimating the infiltration rate parameters, which were finally used in a detailed simulation of the infiltration, leading to its validation with respect to experimental data.

## 2. Materials and Methods
### Experimental Materials and Methods

The infiltration experimental setup was described in previous publications [17–19]. As schematized in Figure 1, the setup features a graphite tube resistively heated by Jthe oule effect and is inserted in a high-pressure Inconel vessel, used as a batch reactor. The experiments were conducted with cylinder-shaped 3D carbon fiber preforms, with an initial open porosity around 77%, drilled in their middle and adjusted around the graphite resistor. Table 1 lists the dimensions of the device. The temperature was measured by a thermocouple located inside the carbon resistor and protected by an alumina tube.

The reactor was filled with methane, and the pressure was maintained at a sufficiently high value to exceed the critical point ($P_c$ = 46 bar; $\theta_c$ = −82 °C) [29], typically between 50 and 70 bar for the initial pressure; the power was varied between 1.5 and 3.0 kW, resulting in temperatures recorded inside the graphite resistor ranging between 900 and 1200 °C. For a better knowledge of the actual temperature inside the fibrous preform, a preliminary thermal study was conducted (run # P1), with several thermocouples inserted in the preform: this allowed establishing a correspondence between the temperature measured in the hollow susceptor and the actual temperatures in the preform.

**Table 1.** Specifications of the experimental setup.

| Part | Material | Dimensions |
|---|---|---|
| Reactor | Inconel | $V_{tank} = 0.3$ L |
| Gas inlet | Inconel | |
| Sealing cap | Inconel | |
| Electrodes | Inconel | |
| Clamps | Steel | |
| Resistor | Graphite tube | $H = 60$ mm ; $d_{ext} = 6$ mm; $d_{int} = 3$ mm |
| Preform | Carbon fibers ($d_f = 7$ μm) | $H = 20$ mm ; $d = 15$ mm |

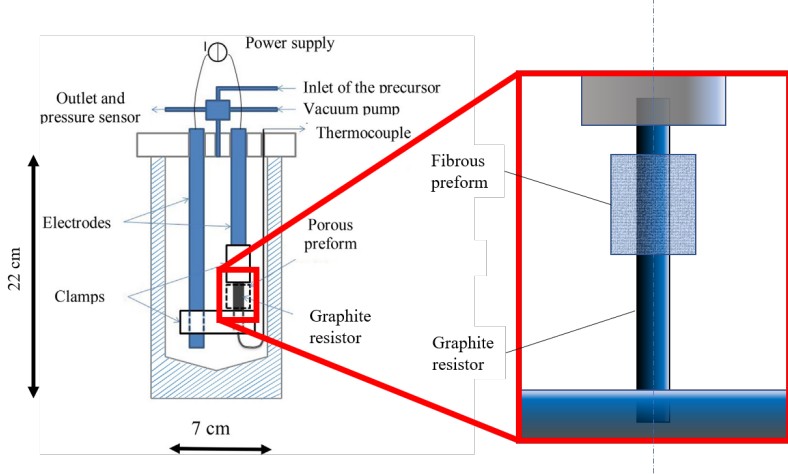

**Figure 1.** Scheme of the setup, with the porous preform settled around the graphite resistor.

During the infiltration runs, the pressure was always increasing due to the global increase in the temperature in the vessel. In these conditions, the gas equation of state was not far from the perfect gas law.

Three runs were achieved for three different initial pressures, with a different power level used for each run. Table 2 summarizes the operating conditions of the $3 \times 3$ working conditions (runs # P2–10).

To measure the deposition profiles, the samples were observed by optical microscopy to study the thickness of carbon deposits around the fibers. After the infiltration process, the samples were cut perpendicularly to the cylinder axis in two halves. The lower halves were then cut along the cylinder axis to study the radial density profiles. All samples were embedded in an epoxy matrix and polished. Some micrographs were processed with ImageJ software [30]. They were first scaled and their brightness and contrast were optimized; then, the images were thresholded to differentiate infiltrated zones from pore space, and the relative area of the solid phase was recorded. Repeating the operation on selected zones along the preform radius allowed constructing radial porosity profiles that could be compared to the simulation results.

A single-filament deposition setup was also designed, built, and tested, and used to determine the rate of chemical deposition of carbon from pure methane. This facility, using the same pressure vessel as described above, had a different top lid that was equipped with a window port allowing direct pyrometric measurement of the temperature on the substrate. Figure 2 depicts the scheme of this setup. The filament was a 30 μm diameter C filament, initially intended to be the core of large CVD-SiC SCS-6 filaments [31], purchased from Textron Specialty Materials (formerly Avco). It had enough electrical conductivity to allow Joule heating up to the desired temperatures. Since it has a very small surface area, the deposition reaction did not significantly alter the composition of the gases of the whole vessel or provoke any significant increase in the reactor pressure, ensuring that the kinetic rate measurements were well-related to the initial pressure of methane only.

**Table 2.** Description of the operating conditions for the infiltration experiments.

| Run # | Initial Pressure (bar) | Power (kW) | Time (min) | Measured Temperature * (°C) |
|---|---|---|---|---|
| P1 | 50 | 1.87 | 15 | ≈850 |
| P2 | 50 | 2.0 | 15 | ≈900 |
| P3 | 50 | 2.0 | 15 | ≈900 |
| | | then 2.5 | 15 | ≈1050 |
| P4 | 50 | 2.0 | 15 | ≈900 |
| | | then 2.5 | 15 | ≈1050 |
| | | then 3.0 | 15 | ≈1200 |
| P5 | 60 | 2.0 | 15 | ≈900 |
| P6 | 60 | 2.0 | 15 | ≈900 |
| | | then 2.5 | 15 | ≈1050 |
| P7 | 60 | 2.0 | 15 | ≈900 |
| | | then 2.5 | 15 | ≈1050 |
| | | then 3.0 | 15 | ≈1200 |
| P8 | 70 | 2.0 | 15 | ≈900 |
| P9 | 70 | 2.0 | 15 | ≈900 |
| | | then 2.5 | 15 | ≈1050 |
| P10 | 70 | 2.0 | 15 | ≈900 |
| | | then 2.5 | 15 | ≈1050 |
| | | then 3.0 | 15 | ≈1200 |

* Temperature was recorded at thermocouple shown in Figure 1, later called TC1.

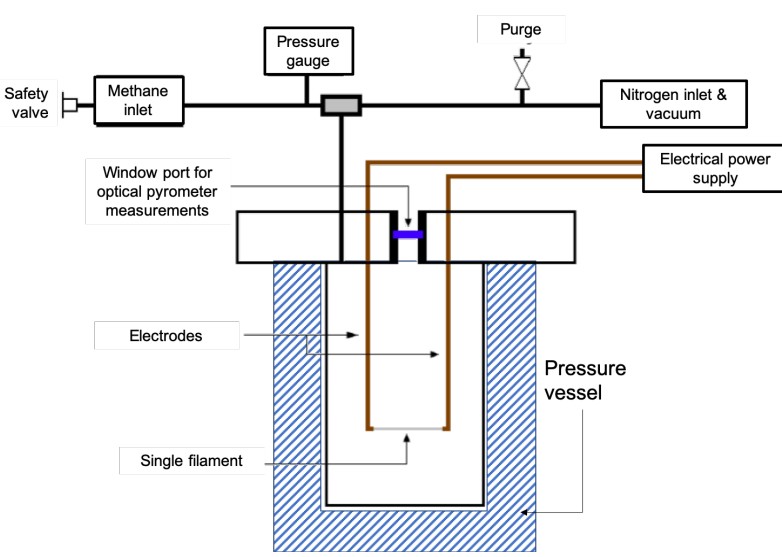

**Figure 2.** Scheme of the setup for the determination of reaction kinetics on a single filament.

## 3. Experimental Results

### 3.1. Kinetics of the Deposition Reaction on a Single Filament

The results from this experimental study are plotted in Figure 3. The deposition rate increased with pressure, except at 50 bar, where it was lower than at 30 bar and comparable temperatures. For pressures of 10 bar and above, homogeneous nucleation was observed at temperatures exceeding a threshold (indicated by closing brackets on the curves) that decreased with increasing pressure, resulting in unreliable measurements unreliable when occurring.

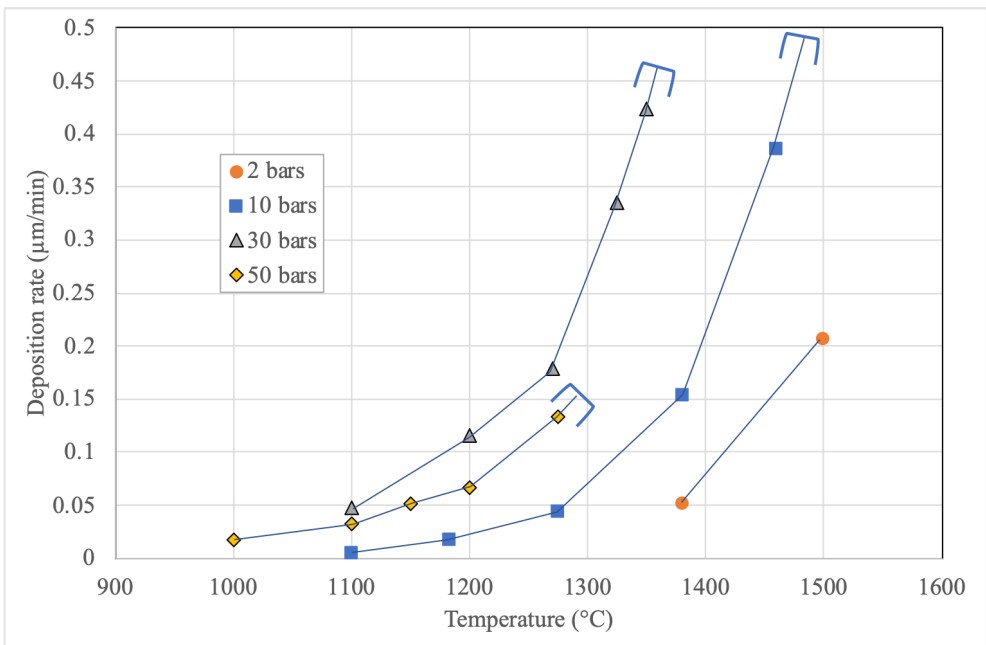

**Figure 3.** Deposition rate on the single filament as a function of temperature and for different methane pressures. The start of the observed homogeneous nucleation is indicated by the brackets ending the tendency curves for 10, 30, and 50 bars.

To interpret the raw data, we had to address the existence of the reverse reaction. Le Châtelier's principle indicates that the reverse reaction (carbon etching by hydrogen, forming methane) will increase in importance with pressure, as is visible from the positive net gaseous mole balance in the balance equation:

$$CH_4(g) \longrightarrow C(s) + 2H_2(g) \tag{1}$$

Hence, the driving force for the deposition reaction becomes relatively lower at high pressures because of a shift in the equilibrium pressures of methane and hydrogen. The equilibrium constant of Equation (1) is given by:

$$K_p = \exp\left(-\frac{\Delta_{dep}G}{\mathcal{R}T}\right) = \frac{P_{h,\text{eq}}^2}{P_{m,\text{eq}}P_{\text{ref}}} \tag{2}$$

where $P_{\text{ref}} = 1$ bar. Assuming that reaction (1) is the only one to take place, the equilibrium pressure of methane is given by:

$$P_{m,\text{eq}} = \frac{P_{\text{ref}}K_p}{4}\left(\sqrt{1 + \frac{4P_{\text{tot}}}{P_{\text{ref}}K_p}} - 1\right)^2 \tag{3}$$

Figure 4 is a plot of the equilibrium methane pressure in the investigated range of pressures and temperatures. It is clearly shown that this equilibrium pressure is far from negligible in many cases considered in this experimental campaign; likewise, the equilibrium mole fraction of methane can also be important at high pressures and relatively low temperatures.

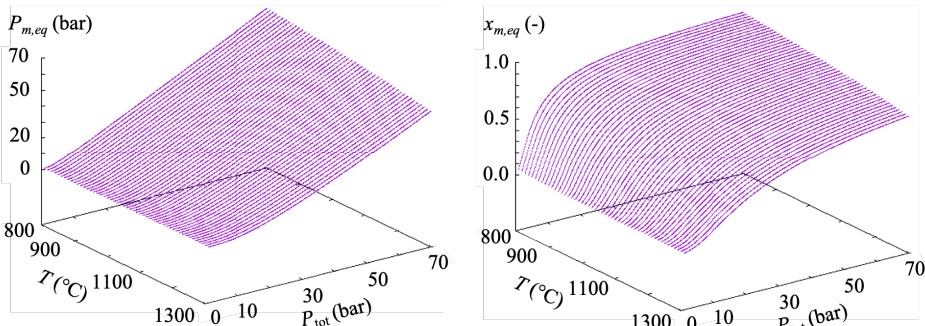

**Figure 4.** Equilibrium partial pressure of methane (**left**) and equilibrium mole fraction of methane (**right**) for the deposition reaction (1) as a function of temperature and total pressure.

One method to quantify the driving force of a reaction is to take the difference between the current methane pressure and its equilibrium value. So, we could interpret the kinetic data with the following rate law:

$$R_{dep} = \underbrace{A \exp\left(\frac{E_a}{\mathcal{R}T}\right)}_{k_{dep}} (P_m - P_{m,\text{eq}}) \tag{4}$$

The rate constant, obtained by dividing the observed rate by the driving force, could then be plotted against temperature in an Arrhenius plot, as shown in Figure 5, for the different investigated pressures. Again, we found that the reaction constant was lower at 50 bar than at 30 bar, clearly indicating a decrease in reaction efficiency when the methane pressure reached the critical pressure. This graph also shows that the activation energy $E_a$ and pre-exponential parameter $A$ varied strongly with total pressure. Table 3 summarizes their values, as also reported in Figure 6. A clear correlation of the pre-exponential factor to the activation energy can be observed.

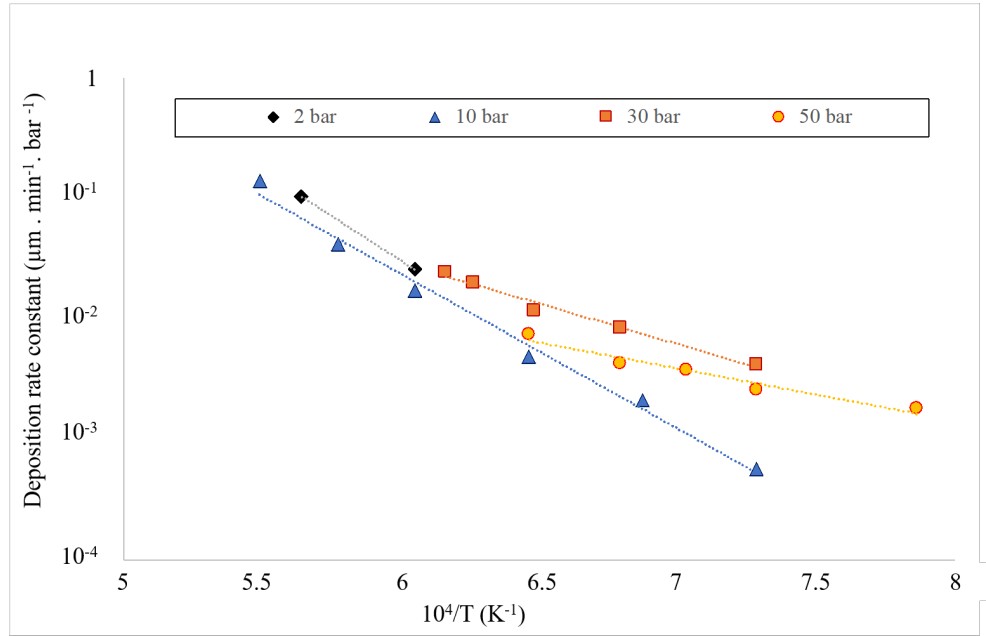

**Figure 5.** Arrhenius plots of the reaction rate for 4 different pressures.

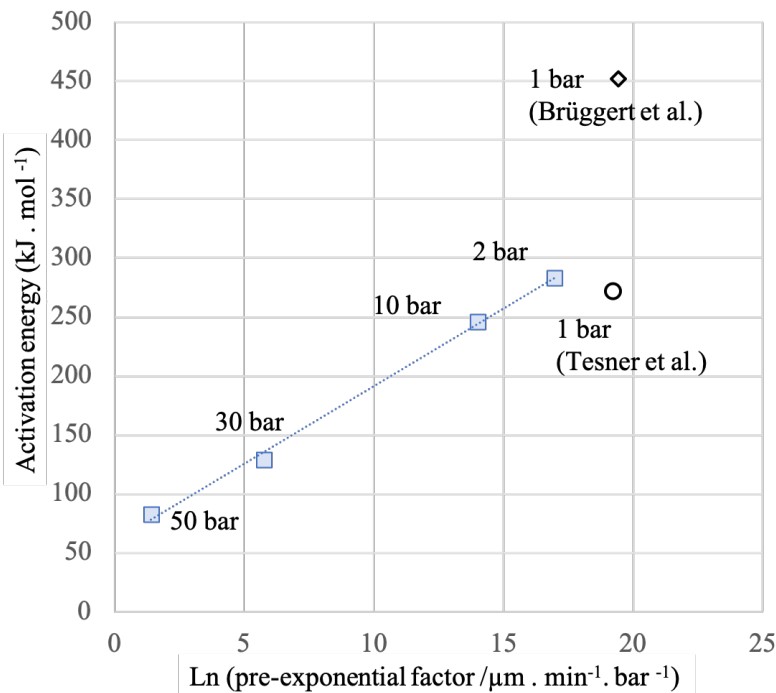

**Figure 6.** Pre-exponential factor—activation energy diagram for methane deposition at various pressures and following various authors. The dotted line is a guide for the eye to the data from this study.

**Table 3.** Arrhenius law parameters obtained at different pressures.

| Total Pressure (bar) | Pre-Exponential Constant $A$ ($\mu m \cdot min^{-1} \cdot bar^{-1}$) | Activation Energy ($kJ \cdot mol^{-1}$) | Ref. |
|---|---|---|---|
| 1 | $2.80 \times 10^8$ | 451.9 | [32] |
| 1 | $2.18 \times 10^8$ | 272 | [33] |
| 2 | $(2.3 \pm 0.2) \times 10^7$ | $284 \pm 6$ | This work |
| 10 | $(1.2 \pm 0.1) \times 10^6$ | $246 \pm 5$ | This work |
| 30 | $(3.2 \pm 0.3) \times 10^2$ | $129 \pm 3$ | This work |
| 50 | $(4.1 \pm 0.4)$ | $83 \pm 2$ | This work |

At relatively low pressures, $E_a$ and ln $A$ were high and almost comparable to values published in the literature at ambient pressure ($\approx$270 kJ·mol$^{-1}$ [33]), in disagreement with other published data [32]. The large discrepancy between the latter two references probably arose from the fact that Tesner et al. [33] produced lumped, overall growth kinetics from methane with an unspecified degree of purity, whereas Brüggert et al. [32] obtained their data from high-purity methane diluted in argon. It is known that the presence of other hydrocarbons, even in small amounts, may strongly accelerate the gas-phase chemistry of methane. It is therefore no surprise that our data better match those of Tesner et al. than those of Brüggert et al..

The pre-exponential factor and activation energy decreased steadily with pressure, with a clear correlation between the two. In principle, an activation energy decrease is expected through the effect of the activation volume:

$$E_a = \Delta U^{\ddagger} + P_{tot}\Delta V^{\ddagger} \tag{5}$$

However, the importance of the decrease in the activation energy with pressure prevents simply invoking this phenomenon; actually, the precise mechanism of carbon deposition is much more complex and involves several steps, the details of which are not accessible

here. One possibility is the appearance of three-body reactions, which can be negligible in atmospheric pressure conditions but strongly enhanced by high pressures.

### 3.2. Thermal Study of an Infiltration Reactor

During infiltrations, it was not possible to have direct access to the substrate temperature, as opposed to the single-filament configuration. Therefore, it was necessary to carry out a preliminary thermal study in order to adequately correlate the temperature read by the control thermocouple and the actual temperatures inside the preform. The specific run #P1 was performed with an initial power of 0.5 kW, a 12 min linear heating ramp, and a power dwell at 1.87 kW. Four extra thermocouples were located in the sample and its surroundings. Figure 7 illustrates the position of the thermocouples and depicts a plot of their thermograms during the heating experiment.

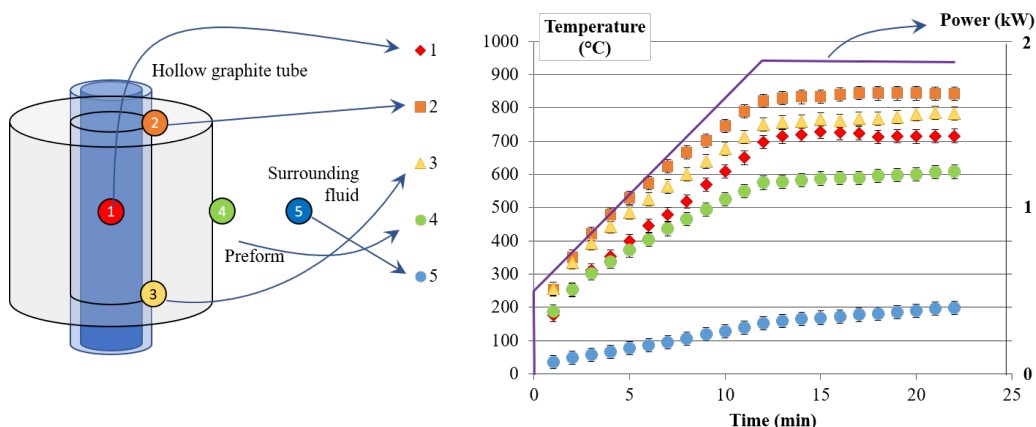

**Figure 7.** Thermal study: (**left**) position of the thermocouples; (**right**) thermal history.

It can be seen that the thermocouples inserted between the preform and the heating tubes were those that recorded the highest temperatures, even more so for the thermocouple located in the top part of the setup. We observed a marked difference between the temperature recorded in the interior of the hollow resistor (which was present in all runs and served as a reference) and the actual temperatures in the porous preform: there was a shift of up to 130 °C for a recorded temperature of 715 °C, i.e., a ≈ 13% absolute temperature excess. Between the top and bottom thermocouples, a thermal difference of 60 °C was also present. Two factors could explain this difference: first, the heat produced in the tube by Joule heating was partially evacuated by conduction in the tube toward the upper and lower clamps, with a better efficiency toward the lower clamp; second, natural convection occurred in the fluid, leading to more intense fluid/preform exchanges in the lower part than in the higher part.

### 3.3. Infiltrations: Kinetics and Densification Profiles

Figure 8 displays micrographs taken at the half-height of the preforms after runs #P2, P3, and P4, with an initial pressure of 50 bar and increasing deposition temperatures. Figure 9 presents the corresponding porosity profiles. It is clearly shown that the infiltration is inside-out, as desired. It also appears that infiltration had a tendency to follow locally the direction of the fibers.

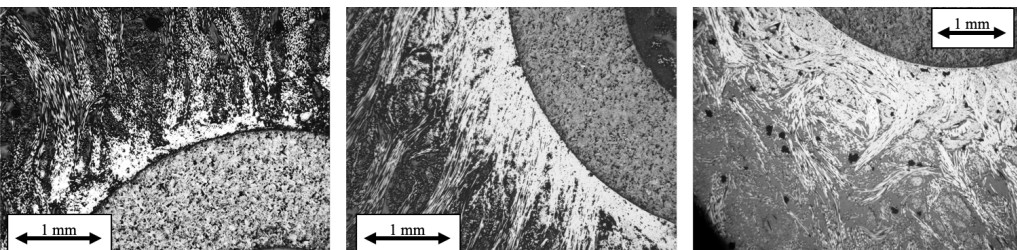

**Figure 8.** Micrographs of the cross-section for 3 successive infiltrations #P2, P3, and P4, from left to right.

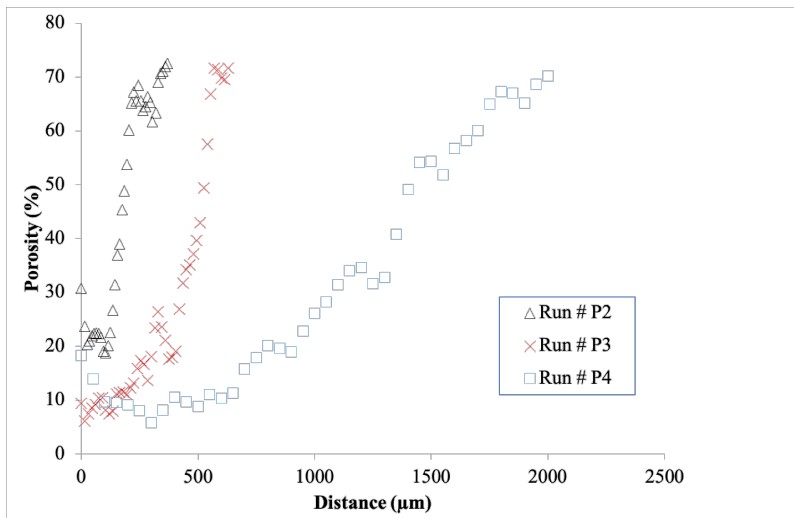

**Figure 9.** Porosity profiles for 3 successive infiltrations #P2, P3, and P4.

Figure 10 compares the infiltrations obtained in all runs from # P2 to # P10 (see Table 2). Clearly, only the 15 min runs performed at 900 °C (see Figure 10a,d,g, left column) only produced faint infiltration, whereas the 45 min runs carried out at three increasingly higher temperatures (900, 1050, and 1200 °C) (see Figure 10c,f,i, right column) led to the presence of a clearly visible infiltrated area with an infiltration front containing a density gradient, while the outer preform had not yet been infiltrated. Additionally, the infiltration front clearly did not progress as much in the lower part than at mid-height. This confirmed that the temperature inside the preform was higher at mid-height than on the top and bottom parts, as shown in the preceding thermal study.

Infiltrations obtained at 60 bar seemed slightly less effective than those at 50 bar; this could be expected from the results of the preceding kinetic study, which evidenced that increasing the pressure over 30 bar had a negative impact on the deposition rate. However, runs carried out at 70 bar pressure seemed to yield a slightly more efficient infiltration. This unexpected result can be attributed to the fact that the homogeneous nucleation of carbon plays a dominant role in these conditions.

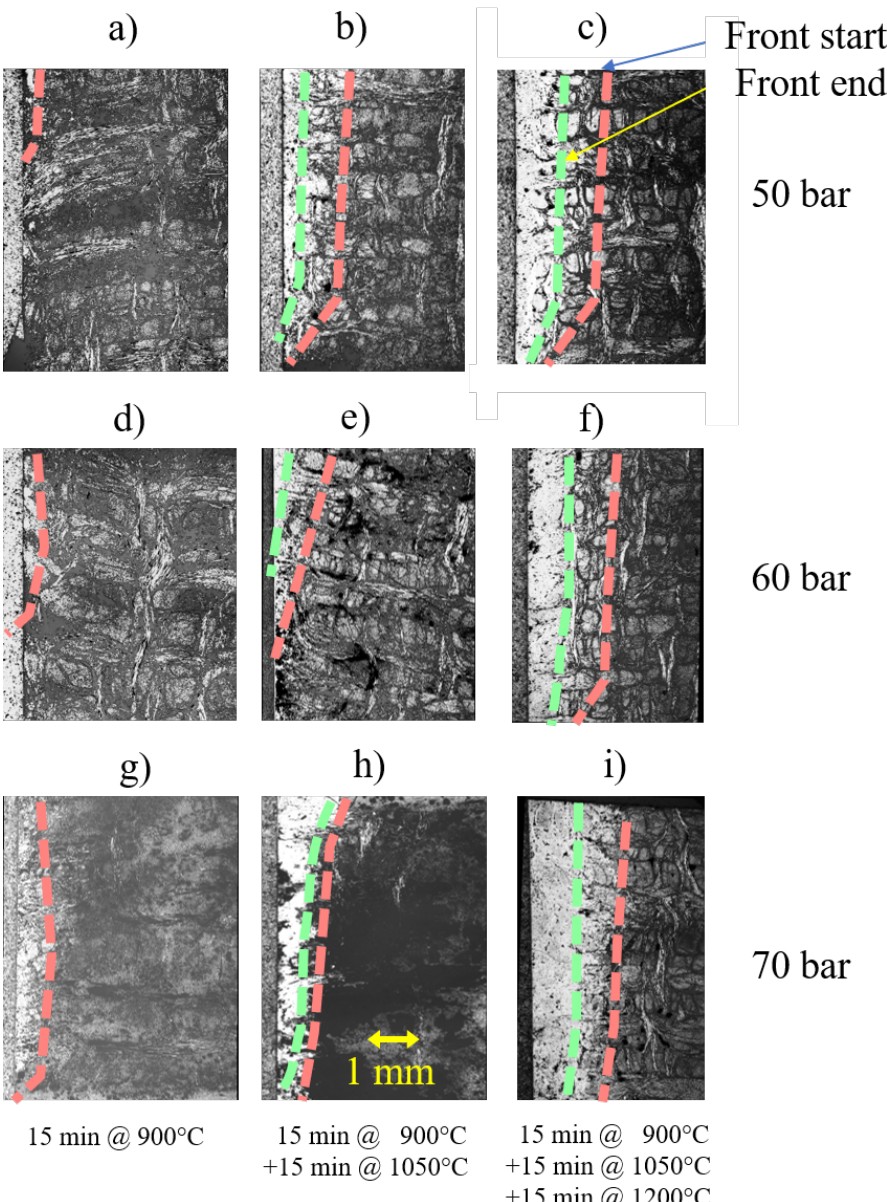

**Figure 10.** (**a**–**i**) Transverse micrographs of samples obtained after runs #P2 to P10, respectively. The scale bar is common to all micrographs. The approximate locations of the front starts and ends are indicated by red and green dashed lines, respectively.

## 4. Infiltration Modeling

*Model Setup*

The modeling approach is an extension of a previous work dedicated to the film-boiling process [23]. Only the densification system with the resistive part and the porous medium was simulated. Considering the cylindrical symmetry, only a radial section was considered; 2D-axisymmetrical geometry was assumed. The resolution domain is described in Figure 11.

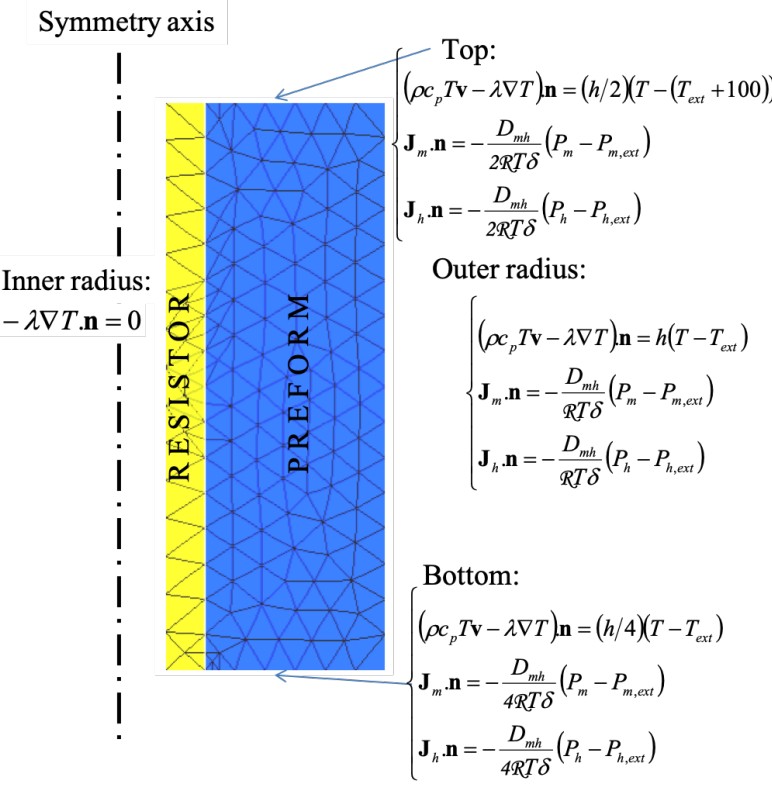

**Figure 11.** Geometry and boundary conditions for the model.

All the physical parameters used for model validation are listed in Table 4 [34]. Additionally, since the working temperature was much higher than the critical temperature, the perfect gas law and Dalton's law of mixtures were assumed to hold. As opposed to our preceding work [23], values for the heat conductivity of the carbon fibers were interpolated from Pradère's measurements [35] on ex-PAN fibers, instead of ex-rayon fibers present in RVC-2000® felts.

The problem solving involved three parts. The first was the resolution of the heat equation:

$$\rho c_p(T)\frac{\partial T}{\partial t} + \rho c_p(T)div \cdot (\mathbf{v}T) + div \cdot (-\lambda(T)\nabla T) = Q_{th} + R\Delta_r H(T) \tag{6}$$

On the left-hand side, the first term is nonstationary and represents the time evolution of temperature; the second corresponds to convection; the third one corresponds to conduction. On the right-hand side are the heat sources, with the first term being the Joule heating power density and the second one describing the energy released by the reaction with rate $R$. Since we were accompanying the evolution of the density in the material, which was quite slow, we chose to neglect the transient term in the above equation, leaving us with:

$$\rho c_p(T)div \cdot (\mathbf{v}T) + div \cdot (-\lambda(T)\nabla T) = Q_{th} + R\Delta_r H(T) \tag{7}$$

In order to obtain the velocity $\mathbf{v}$ and the reaction source term $R$, it was necessary to solve the momentum, mass, and species balances, which are strongly coupled in a porous medium. Here, we considered only two species, the precursor (methane) and hydrogen, ignoring by-products. We obtain:

$$\begin{cases} \frac{\partial}{\partial t}\left(\frac{P_m M_m}{\mathcal{R}T}\right) + div \cdot \mathbf{J}_m &= -R \\ \frac{\partial}{\partial t}\left(\frac{P_h M_h}{\mathcal{R}T}\right) + div \cdot \mathbf{J}_h &= 2R \end{cases} \tag{8}$$

**Table 4.** Physical parameters employed in the numerical model.

| Parameter | Value or Expression | Unit |
|---|---|---|
| Reactor | | |
| Initial outer wall temperature | $T_{ext} = 298$ | K |
| Initial total pressure | $P_{ext} = 5,\ 6,\ \text{or}\ 7 \times 10^6$ | Pa |
| Diffusion boundary layer thickness | $\delta = 0.5$ | m |
| Heat capacity of the reactor | $m_{tank}c_{p,tank} = 300$ | J·K$^{-1}$ |
| Reactor volume | $V_{tank} = 3 \times 10^{-4}$ | m$^3$ |
| Outer heat exchange coefficient | $h_w S_w = 10$ | W·K$^{-1}$ |
| Preform/fluid heat transfer coefficient | $h = 4200$ | W·m$^{-2}$·K$^{-1}$ |
| Resistor/exterior heat transfer coefficient | $h_{res} = 6000$ | W·m$^{-2}$·K$^{-1}$ |
| Preform | | |
| Fiber density | $\rho_f = 1840$ | kg·m$^{-3}$ |
| Fiber initial diameter | $d_{f,0} = 7 \times 10^{-6}$ | m |
| Initial porosity | $\varepsilon_0 = 0.77$ | - |
| Mass transfer parameters | | |
| Carbon density | $\rho_c = 2180$ | kg·m$^{-3}$ |
| Carbon molar volume | $\Omega_c = 5 \times 10^{-6}$ | m$^3$·mol$^{-1}$ |
| Internal surface area | $\sigma_v = \dfrac{4}{d_f}\left[(2 - \varepsilon_0)\left(\dfrac{\varepsilon}{\varepsilon_0}\right) - \left(\dfrac{\varepsilon}{\varepsilon_0}\right)^2\right]$ | m$^{-1}$ |
| Effective pore diameter | $d_p(\varepsilon) = 4\dfrac{\varepsilon}{\sigma_v}$ | m |
| Viscous flow tortuosity | $\eta_v(\varepsilon) = 2.76\varepsilon^{-2/3}(\ln \varepsilon)^2$ | - |
| Darcy Permeability | $K(\varepsilon) = \varepsilon\dfrac{d_p^2}{32\eta_v}$ | m$^2$ |
| Diffusion tortuosity | $\eta_d(\varepsilon) = \varepsilon^{-2/3}$ | - |
| Mutual diffusion coefficient | $D_{mh} = \dfrac{\varepsilon}{\eta_b}\dfrac{2.62 \times 10^{-8}T^{3/2}}{P_{tot}\sqrt{M_{12}}\sigma_{12}^2\Omega_d}$ | m$^2$·s$^{-1}$ |
| Fluid dynamic viscosity | $\mu = 1.876 + 0.2441T - 4 \times 10^{-5} \cdot T^2$ | Pa·s |
| Heat transfer parameters | | |
| Conductivity, effective | $\lambda_{eff} = (1 - \varepsilon_0)\lambda_f + (\varepsilon_0 - \varepsilon)\lambda_d + \varepsilon\lambda_g$ | W·m$^{-1}$·K$^{-1}$ |
| Conductivity, fibers | $\lambda_f(T) = -25.671 + 0.22728\,T - 1.3159 \times 10^{-4}\,T^2 + 2.4971 \times 10^{-8}\,T^3$ | W·m$^{-1}$·K$^{-1}$ |
| Conductivity, deposit | $\lambda_d(T) = -3.466 + 0.0271\,T - 2.05 \times 10^{-5}\,T^2 + 5.3 \times 10^{-9}\,T^3$ | W·m$^{-1}$·K$^{-1}$ |
| Conductivity, gas | $\lambda_g(T) = -0.02329 + 1.1092 \times 10^{-4}\,T - 2.0 \times 10^{-8}\,T^2$ | W·m$^{-1}$·K$^{-1}$ |
| Molar heat capacities | $c_f = c_d = -42.468 + 2.852T + 0.001T^2$ | J·K$^{-1}$·mol$^{-1}$ |
| | $c_m = 24.38 + 3.3 \times 10^{-2}T + 3.0 \times 10^{-5}T^2 - 2.0 \times 10^{-8}T^3$ | J·K$^{-1}$·mol$^{-1}$ |

The flows of both species contain diffusion terms and a convection term obtained by Darcy's law of viscous transfer in porous media. Since we checked that the pore Knudsen number was always very small ($\approx 10^{-5} - 10^{-4}$), we can write:

$$\begin{cases} \mathcal{R}T\mathbf{J}_m & = & -D_{mh}\nabla P_m + \mathbf{v}P_m \\ \mathcal{R}T\mathbf{J}_h & = & -D_{mh}\nabla P_h + \mathbf{v}P_h \end{cases} \tag{9}$$

where

$$\mathbf{v} = -\mu^{-1}K\nabla(P_m + P_h) \tag{10}$$

and the diffusion coefficients are approximately given by the formulas in Table 4. The precursor consumption molar rate, using Equation (4), is given by:

$$R = \sigma_v(\varepsilon)A\exp\left(-\frac{E_a}{\mathcal{R}T}\right)\left(\frac{P_m - P_{m,eq}}{\mathcal{R}T}\right) \tag{11}$$

where the internal surface $\sigma_v$, itself a function of porosity, is an important parameter (see Table 4). To obtain the problem specification, the infiltration equation has to be given as a scaled solid mass balance:

$$-\frac{\partial \varepsilon}{\partial t} = \Omega_c R \tag{12}$$

where the solid molar volume $\Omega_c$ equals $M_c/\rho_c$. The details of all coefficients and values are reported in Table 4.

The boundary conditions for Equations (7) and (8) are illustrated in Figure 11. For the heat balance equation, we have Fourier boundary conditions, simulating an exchange with the surrounding medium. The heat exchange coefficient is smaller on the upper and lower parts of the domain because the nearby presence of holders hampers the development of convective exchange. Conversely, on the tube, the heat transfer coefficient is higher because of the strong conductive losses toward the upper and lower clamps.

For mass transfer, the boundary conditions are similar to the Fourier boundary conditions. They simulate the species exchange with the surrounding fluid:

$$\begin{cases} \mathcal{R}T\mathbf{J}_m & = & -\frac{D_{mh}}{\delta}(P_m - P_{m,ext}) \\ \mathcal{R}T\mathbf{J}_h & = & -\frac{D_{mh}}{\delta}(P_h - P_{h,ext}) \end{cases} \tag{13}$$

Here, $\delta$ is a given boundary layer thickness, and $P_{i,ext}$ ($i = m, h$) are the external values of the partial pressures, i.e., average values for the fluid lying in the reactor outside the preform. For the same reason as state above, the species transfer coefficient abated on the top and bottom surfaces of the domain. Figure 11 shows the details of these boundary conditions.

During process operation, the surrounding fluid temperature was subject to a constant rise, because the heat transfer from the vessel walls to the ambient was low. This was modeled as an evolution equation for the surrounding fluid temperature $T_{ext}$:

$$m_{tank}c_{p,tank}\frac{dT_{ext}}{dt} = \oiint_{\partial\Omega} \mathbf{q} \cdot \mathbf{n}dS - h_w S_w(T_{ext} - T_{amb}) \tag{14}$$

where $m_{tank}c_{p,tank}$ is the heat capacity of the vessel, $\mathbf{n}$ is the normal exiting the boundary, $h_w S_w$ defines the importance of the thermal exchange with the ambient temperature, and the integral term describes the heat losses of the boundaries of the preform. The external partial pressures also varied throughout the process, because of the increases in temperature and the total number of gas moles. To account for these effects, the following evolution equations were solved together with the preceding ones:

$$\frac{dP_{i,ext}}{dt} = \frac{P_{i,ext}}{T_{ext}}\frac{dT_{ext}}{dt} + \frac{\mathcal{R}T_{ext}}{V_{tank}}\oiint_{\partial\Omega} \mathbf{J}_i \cdot \mathbf{n}dS \tag{15}$$

where the integral term is the total flux of species $i = m$ or $h$ exiting the preform and therefore entering the reactor vessel, with volume $V_{tank}$.

In the experimental setup, the heating power was not instantly set to the desired value; so, the model reproduced the progressive increase in the power (around 125 W·min$^{-1}$) by giving a time-dependent heat source term $Q_{th}$ in Equation (7), only active in the hollow graphite resistor.

The resolution by cubic finite elements was performed using commercial multi-physics software FlexPDE v.7.19 Lite [36].

## 5. Numerical Results and Discussion

### 5.1. Thermal Study

The simulation parameters for heat transfer (essentially the power density and the boundary heat transfer coefficients) were tuned to obtain a satisfactory agreement between the thermograms of thermocouples TC2, TC3, and TC4 described in Figure 7. Figure 12 illustrates this correct agreement. It can be noted that the temperature at the mid-height of the preform/tube interface was markedly higher than at the locations of thermocouples 2 and 3 (up to a 300 K difference).

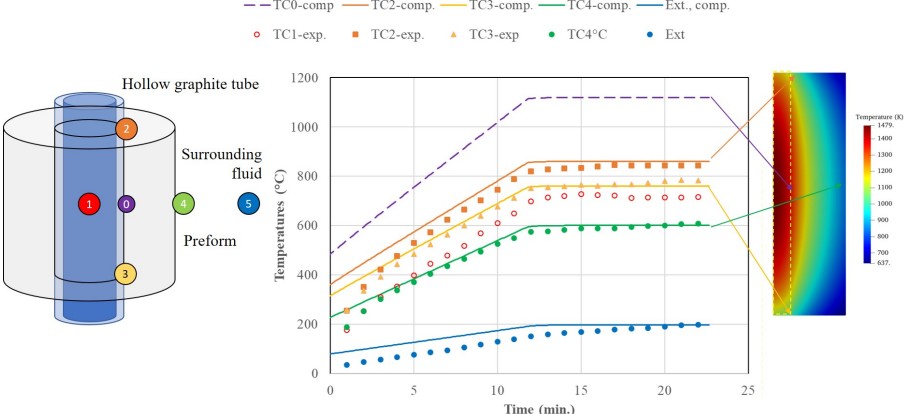

**Figure 12.** Computation of the thermal histories of selected points of the setup: (**left**) position of the sampled points; (**right**) thermal history of these points compared to experimental ones.

A small drift in the temperature occurred at thermocouples TC2, TC3 and TC4 even when the power was held constant, because of the increase in fluid temperature outside of the preform. Moreover, as shown in Figure 13, the temperature at TC4 started increasing appreciably for the highest powers, because infiltration increased the thermal conduction flux between the resistor and its location.

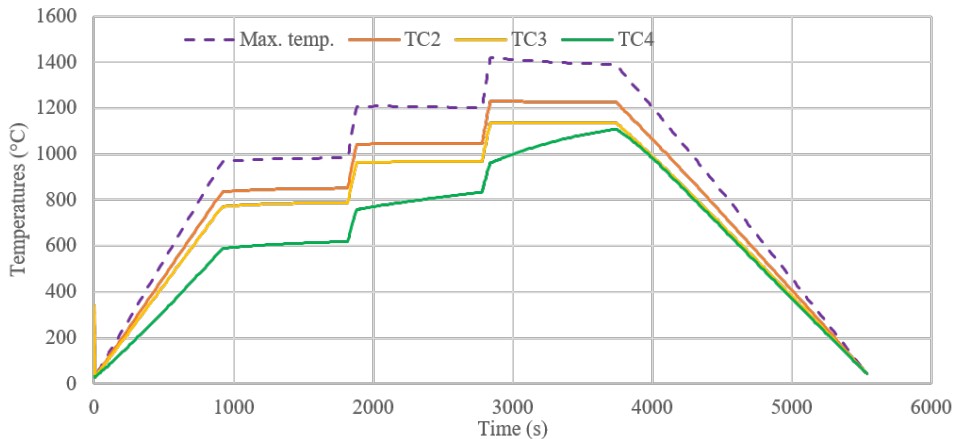

**Figure 13.** Full simulated thermograms of experiment #P4.

The temperatures computed at different measurement points for all runs are reported in Table 5, as well as the temperature measured in the center of the hollow resistor (TC1). It can be seen that an appreciable difference existed between the measured temperature and the highest temperature resulting from the simulations (up to 200 K). No appreciable influence of the total pressure on these temperature values was found, either computationally or experimentally.

**Table 5.** Measured and computed temperatures for different runs.

| Run # | Power kW | TC1 °C | TC2 °C | TC3 °C | TC4 °C | Max. Temp. °C | Radial Flux MW·m$^{-2}$ |
|---|---|---|---|---|---|---|---|
| P1 | 1.87 | 715 | 843 | 783 | 608 | 1113 | 4.94 |
| P2, P5, P8 | 2 | 950 | 922 | 854 | 638 | 1154 | 5.31 |
| P3, P6, P9 | 2.5 | 1025 | 1124 | 1033 | 778 | 1409 | 6.64 |
| P4, P7, P10 | 3 | 1200 | 1334 | 1245 | 945 | 1425 | 7.97 |

*5.2. Evolution of Pressure during the Infiltration Runs*

The total pressure of the vessel was subjected to a strong increase during the run, because of the temperature increase at constant total volume. Figure 14 depictsw the evolution of the total pressure and the methane and hydrogen partial pressures during runs #P2–4. The pressure nearly doubled at the highest heating power. This phenomenon is extremely important to notice because the kinetics and thermodynamics of the deposition reaction are dependent on the total pressure, as shown in the preceding sections. Conversely, the pressure increase due to excess gas mole creation was negligible.

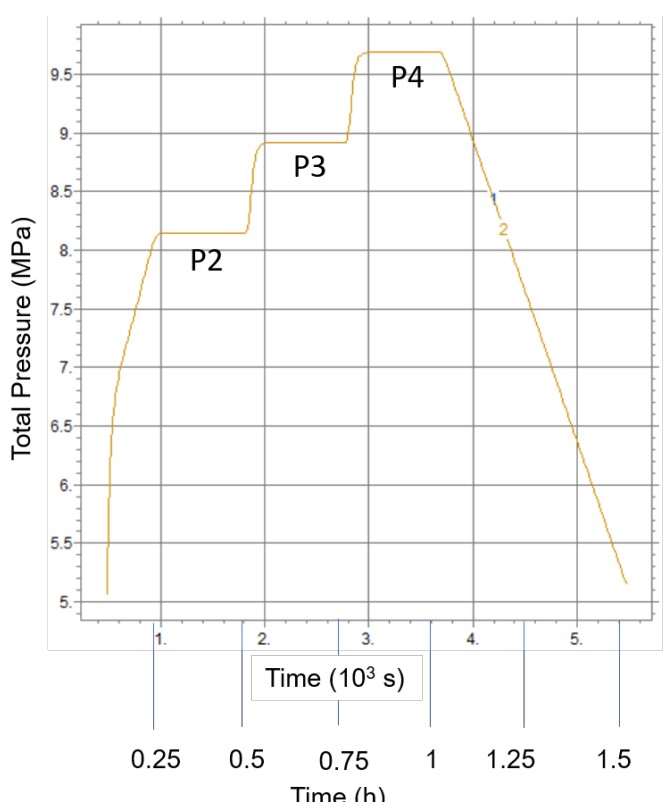

**Figure 14.** Simulated evolution of the total pressure for experiments #P2 to P4.

### 5.3. Infiltration Fronts

Directly modeling the infiltration fronts was not feasible with the kinetic data obtained in the filament CVD conditions. Thee highest total pressure obtained in that system was only 30 bar, while the infiltration runs were mostly carried out at local pressures exceeding 80 bar. Extrapolation of the kinetic data obtained previously would provide irrelevant values of the activation energy and pre-exponential factor. However, it was possible to identify plausible values by using the measurements of the observed front thicknesses and positions.

Simple estimates of the front width and velocity were derived in a previous study [25]. They can be written in the following form:

$$\ell_{front} \approx \frac{\lambda_s T_h}{q} \frac{\mathcal{R} T_h}{E_a} \tag{16}$$

$$v_{front} \approx \ell_{front} A k_{dep}(T_h) \frac{M_s}{\rho_s} \frac{\mathcal{R} T_h}{P_m} \tag{17}$$

where $T_h$ is the hot-side temperature of the front, $\lambda_s$ is the solid-phase conductivity, and $A \approx \frac{4}{d_f}$ is the internal surface area scale parameter. Reverting these equations, estimates of $E_a$ and $k(T_h)$ are obtained as:

$$E_a \approx \frac{\lambda_s \mathcal{R} T_h^2}{q \ell_{front}} \tag{18}$$

$$k_{dep}(T_h) \approx \frac{v_{front} \rho_s (P_m - P_{m,eq})}{\ell_{front} A M_s \mathcal{R} T_h} \tag{19}$$

Here, we took $(P_m - P_{m,eq})$ as the driving force of the reaction instead of $P_m$, as suggested in Equation (4).

The values obtained applying these formulas to cases #P2–P10 (Figure 10) are summarized in Table 6. Figure 15 depicts the kinetic data obtained from this table and from the single-filament CVD experiments. It is shown that the activation energy decrease, marked in the 1–50 bar range, did not continue in the high-pressure infiltration experiments; $E_a$ was roughly in the 60–80 kJ·mol$^{-1}$ interval. The pre-exponential factor did not show any clear evolution at high pressures and remained in the [3; 6] interval for its logarithm. This saturation effect can be attributed to the fact that, even though the total pressure was higher in the infiltration experiments, the driving force $P_m - P_{m,eq}$ varied modestly. Of course, these estimates are rough, but the consistency between the single-filament CVD data and preform infiltration data is evidenced.

**Table 6.** Analysis of runs #P2–P10. Temperatures and pressures were obtained from the thermal modeling. Front width and velocity values were extracted from Figure 10 when possible. Activation energies and pre-exponential factors were obtained from the previous data and Equations (18) and (19).

| Run # | $P_{tot}$ bar | $P_m - P_{m,eq}$ bar | $\ell_{front}$ mm | $x_{front}$ mm | $v_{front}$ nm·s$^{-1}$ | $E_a$ kJ·mol$^{-1}$ | ln($A$) µm·min$^{-1}$·bar$^{-1}$ |
|---|---|---|---|---|---|---|---|
| P2 | 81 | 14.72 | | | | | |
| P3 | 89 | 24.45 | 1.2 | 1.4 | 1.56 | 80 | 5.35 |
| P4 | 96 | 25.35 | 1.0 | 1.95 | 0.61 | 81 | 4.09 |
| P5 | 98 | 14.97 | | | | | |
| P6 | 107 | 25.11 | 1.55 | 1.45 | 1.6 | 62 | 3.91 |
| P7 | 116 | 26.02 | 1.15 | 1.85 | 0.45 | 71 | 2.92 |
| P8 | 114 | 15.14 | 1.2 | 1.1 | 1.22 | 72 | 5.14 |
| P9 | 125 | 25.59 | 0.9 | 1.2 | 1.11 | 106 | 4.48 |
| P10 | 136 | 26.50 | 1.1 | 2.45 | 1.39 | 74 | 3.86 |

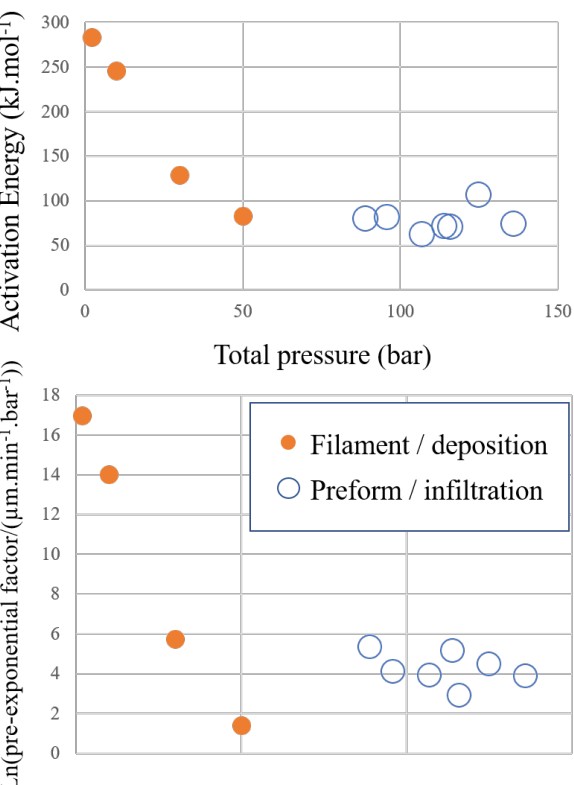

**Figure 15.** Kinetic parameters *vs.* total pressure in CVD and CVI experiments. The size of the symbols indicates the experimental uncertainty.

Using these estimated values of the kinetic parameters, the minimal heat flux for the existence of an infiltration front could be estimated. The relation is [25] :

$$q >\approx \frac{\lambda_s \mathcal{R} T_h^2}{E_a} \sqrt{\frac{A k_{dep}(T_h)}{D_m}} \qquad (20)$$

The obtained critical fluxes were consistently between 10 and 30 times less than the actual heat fluxes, confirming the consistency of the criterion with experimental data.

Finally, an infiltration run was simulated in detail using the FE solver for runs #P2–4, using an activation energy of 70 kJ·mol$^{-1}$ and a pre-exponential factor of $\exp(4.5)$ μm·min$^{-1}$·bar$^{-1}$. The results are displayed in Figure 16, showing a very satisfactory agreement between the computed and experimental density profiles. In particular, the shape of the infiltration front, principally dictated by the thermal field, was correctly reproduced. These results further validate the modeling approach applied in this study.

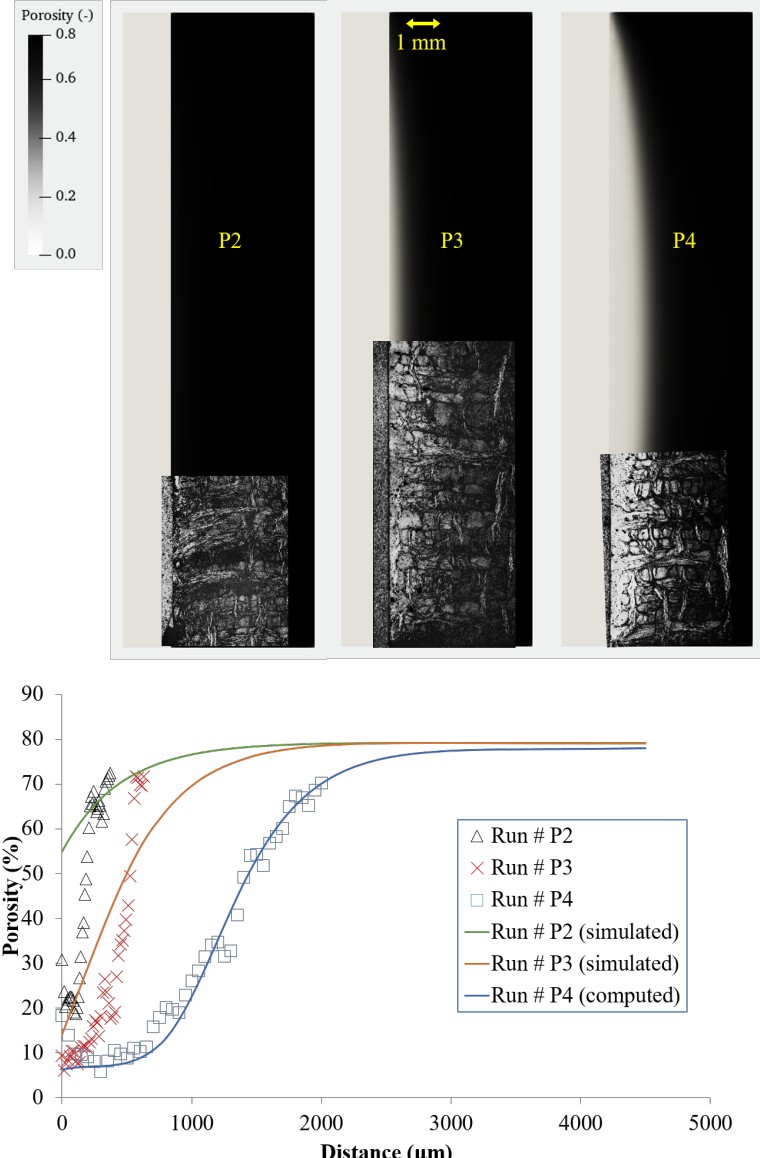

**Figure 16.** Simulated infiltration runs #P2–4: porosity profiles: (**top**) 2D computed profiles compared to micrographs on the same scale; (**bottom**) comparison of the porosity profiles obtained at the half-height of the preform.

## 6. Summary and Outlook

In this work, we considered the chemical vapor deposition and infiltration of pyrolytic carbon at high pressures, well above the critical point of methane. Kinetic data obtained from single-filament CVD evidenced that the driving force for deposition is not the methane partial pressure, but rather its difference with its equilibrium partial pressure with carbon and hydrogen. The reversibility of the deposition reaction becomes important at very high pressures due to Le Châtelier's principle. If temperature and pressure continue to increase, we would eventually reach a situation in which carbon would be attacked by hydrogen to produce methane. In the infiltration experiments, we clearly observed that there was no point in increasing the initial total pressure above 50 bar, since the gain in infiltration rate was modest.

The infiltration results were interpreted through an FE simulation of heat and mass transfer, allowing a correct reconstitution of the actual temperature and pressure conditions experienced by the preform during the infiltration runs. The steep thermal gradients evidenced were high enough to favor the presence of an infiltration front progressing

from the inside to the outside of the preform. The infiltration front analysis yielded rough estimates of the kinetic parameters in line with the CVD experiments (which were performed at lower pressures). All data showed that the apparent activation energy strongly decreased from the atmospheric pressure down to $\approx$50 bar, then remained constant around 70 kJ·mol$^{-1}$. A detailed modeling of the infiltration confirmed the estimates made with the simple formulas. Despite the low value of the activation energy, infiltration fronts were obtained, which is a favorable situation that yields optimal infiltration quality, that is, the lowest possible residual porosity.

This physico-chemical study provided a sufficient data set and a modeling framework for the possible upscaling of the high-pressure thermal-gradient CVI process. The model results can be used for the research of an optimal set of parameters for the fastest and cheapest possible infiltrations. For instance, an interesting perspective is to study the potential of a continuously fed reactor that may overcome some difficulties encountered in these batch reactor experiments, such as the impossibility of completing infiltration in a single run.

**Author Contributions:** Conceptualization, L.M., R.P., Y.L.P. and A.G.; methodology, L.M., R.P., Y.L.P. and G.L.V.; software, G.L.V.; validation, G.L.V.; formal analysis, G.L.V.; investigation, G.T., Q.B., L.M. and G.L.V.; resources, L.M. and G.L.V.; data curation, G.T., Q.B., L.M. and G.L.V.; writing—original draft preparation, G.L.V.; writing—review and editing, L.M.; visualization, G.L.V.; supervision, G.L.V., R.P., L.M. and A.G.; project administration, L.M. and A.G.; funding acquisition, A.G. All authors have read and agreed to the published version of the manuscript.

**Funding:** This work was funded by Herakles (now Safran Ceramics) through masters internship stipends to G.T. and Q.B.

**Data Availability Statement:** Experimental and numerical data supporting this paper are available upon request from G.L.V.

**Acknowledgments:** The authors wish to acknowledge J. Roger (Bordeaux University, LCTS) for his help in providing thermochemical calculations, and S. Couthures (CNRS, LCTS) for his technical assistance.

**Conflicts of Interest:** The authors declare no conflict of interest.

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
