# Peer review of "Chemical Supercritical Fluid Infiltration of Pyrocarbon with Thermal Gradients: Deposition Kinetics and Multiphysics Modeling"

_jcs, doi:10.3390/jcs6010020_

Round 1

Reviewer 1 Report

The paper describes a coupled experimental/numerical simulation study of deposition kinetics of carbon during chemical supercritical infiltration. The experiments have been cleverly designed to determine, first, the reaction kinetics and, separately, the infiltration kinetics.  A comprehensive numerical framework with all pertinent reactor conditions is used to simulate the infiltration process. An interesting result (though perhaps unsurprising once one sees it) is the attainment of an optimal pressure and temperature for the process. The paper is generally well-written and thorough. It could be published in its current form.

A few very minor suggestions:

  1. Increase the size of the axes labels in Figure 4 (they are almost imperceptible to my eyes).
  2. Provide estimates of the uncertainties of the pre-exponential constants and activation energies in Table 3.
  3. Use a different color for the scale bar and increase font size on the micrographs in Figure 8.
  4. On page 15, line 229, there is a missing closing parenthesis, ), after “…heat transfer coefficients”.

Overall, a very fine piece of work!

Author Response

Reviewer #1 : The paper describes a coupled experimental/numerical simulation study of deposition kinetics of carbon during chemical supercritical infiltration. The experiments have been cleverly designed to determine, first, the reaction kinetics and, separately, the infiltration kinetics.  A comprehensive numerical framework with all pertinent reactor conditions is used to simulate the infiltration process. An interesting result (though perhaps unsurprising once one sees it) is the attainment of an optimal pressure and temperature for the process. The paper is generally well-written and thorough. It could be published in its current form.

Many thanks for this very positive assessment.

A few very minor suggestions:

  1. Increase the size of the axes labels in Figure 4 (they are almost imperceptible to my eyes).

Done.

  1. Provide estimates of the uncertainties of the pre-exponential constants and activation energies in Table 3.

Done.

  1. Use a different color for the scale bar and increase font size on the micrographs in Figure 8.

Done.

  1. On page 15, line 229, there is a missing closing parenthesis, ), after “…heat transfer coefficients”.

Done.

Overall, a very fine piece of work!

Reviewer 2 Report

Accept

Author Response

Thank you for your positive recommendation. Concerning the English we have tried to improve the language as much as possible.

Reviewer 3 Report

A relatively new process, chemical supercritical superfluid infiltration for materials processing was introduced and used for composites manufacturing. Both process demonstration and numerical study were presented. The following corrections may be made for publishing.

  1. On lines 60-61, "insight on the process' May be corrected as "insight into the process."
  2. On line 69, "for estimates of the infiltration rate parameters" should be "for estimating the infiltration rate parameters" or "for the estimation on the infiltration rate parameters".
  3. In Figure 1, what is the material for making the clamps?
  4. Referring to both Fig. 1 and Fig. 2, a little bit more details in the facilities may be given. For example, what is the material for the electrode? What is the material for the sealing cap to hold the window port and the electrodes and the inlet for the infusion gas? This information may helpful for readers to repeat the experiment of infiltration.
  5. Is the reaction under continuous gas flowing condition or not? It is not clear from the facility set up.
  6. Redraw Figure 5 by putting 1/T at the bottom instead of on the top of the figure.

Author Response

Reviewer #3 : A relatively new process, chemical supercritical superfluid infiltration for materials processing was introduced and used for composites manufacturing. Both process demonstration and numerical study were presented. The following corrections may be made for publishing.

  1. On lines 60-61, "insight on the process' May be corrected as "insight into the process."

Done.

  1. On line 69, "for estimates of the infiltration rate parameters" should be "for estimating the infiltration rate parameters" or "for the estimation on the infiltration rate parameters".

Done.

  1. In Figure 1, what is the material for making the clamps?

The clamps were made of steel. This is now indicated in Table 1.

  1. Referring to both Fig. 1 and Fig. 2, a little bit more details in the facilities may be given. For example, what is the material for the electrode? What is the material for the sealing cap to hold the window port and the electrodes and the inlet for the infusion gas? This information may helpful for readers to repeat the experiment of infiltration.

The electrodes, reactor body and sealing cap were made of Inconel alloy; the resistor tube was made of graphite. These pieces of information are now more clearly indicated in Table 1.

  1. Is the reaction under continuous gas flowing condition or not? It is not clear from the facility set up.

The reactor is a batch reactor. This was already mentioned in the conclusion and is now indicated in the text p. 2, line 76.

  1. Redraw Figure 5 by putting 1/T at the bottom instead of on the top of the figure.

Done.